# Effects of Instability Neuromuscular Training Using an Inertial Load of Water on the Balance Ability of Healthy Older Women: A Randomized Clinical Trial

**DOI:** 10.3390/jfmk9010050

**Published:** 2024-03-13

**Authors:** Shuho Kang, Ilbong Park

**Affiliations:** Department of Sports Rehabilitation, Busan University of Foreign Studies, Busan 46234, Republic of Korea; 20236204@bufs.ac.kr

**Keywords:** dynamic stability, inertial load of water, instability neuromuscular training, older women, balance ability

## Abstract

The reflexive responses to resist external forces and maintain posture result from the coordination between the vestibular system, muscle, tendon, and joint proprioceptors, and vision. Aging deteriorates these crucial functions, increasing the risk of falls. This study aimed to verify whether a training program with water bags, an Instability Neuromuscular training (INT) using the inertial load of water, could positively impact balance ability and dynamic stability. This study was conducted with twenty-two healthy older women aged ≥ 65 (mean age: 74.82 ± 7.00 years, height: 154.20 ± 5.49 cm, weight: 55.84 ± 7.46 kg, BMI: 23.55 ± 3.58 kg/m^2^). The participants were randomly allocated into two groups: a group that used water bags and a control group performing bodyweight exercises. The intervention training lasted 12 weeks, with 2 sessions per week totaling 24 sessions, each lasting 60 min. The pre- and post-tests were compared using *t*-tests to examine within- and-between-group differences. The effect size was examined based on the interaction between group and time using a two-way repeated measures ANOVA. The Modified Timed Up and Go manual (TUG manual), Sharpened Romberg Test (SRT), and Y-balance test (YBT) were conducted to assess dynamic stability, including gait function, static stability, and reactive ability. In comparison between groups, the waterbag training group showed a decrease in task completion time associated with an increase in walking speed in the TUG manual test (*p* < 0.05), and an increase in static stability and reaction time in the Sharpened Romberg test with eyes closed (*p* < 0.05), and an increase in single-leg stance ability in both legs in the Y-balance test (*p* < 0.05). All statistical confidence interval levels were set 95%. INT using the inertial load of water enhanced the somatosensory system and gait related to dynamic stability in older women. Therefore, the clinical application of this training program is expected to reduce the risk of falls in healthy older women, improving dynamic stability related to gait.

## 1. Introduction

Balance ability refers to the ability of the body to adjust its posture or movement quickly under specific conditions and is the reflexive action involving the vestibular system, muscles and tendons, joint proprioceptors, and visual coordination to maintain posture against external forces [1]. Dynamic balance in particular is crucial for maintaining, transitioning, and stabilizing various postures [1], and, for older adults aged 65 and older, falls resulting from deterioration in dynamic balance ability are a leading cause of injury directly linked to death [2].

Falls are defined as unexpected transitions from a higher to a lower position or the ground, typically resulting from sudden changes in posture [3]. Particularly, 10% to 20% of falls in older adults result in serious injuries such as fractures or head traumas, and non-fatal fall injuries are related with considerable morbidity, including decreased functioning and loss of independence [4]. After experiencing a fall, there is a high risk of secondary conditions, such as severe fear, decreased functionality, and mental anxieties related to depression [5]. For women, catabolic hormone activity is higher than in men, resulting in greater muscle mass loss [6]. Furthermore, women are more susceptible to fractures due to osteoporosis, caused by a decrease in bone density after menopause [7]. Balance training, as well as strength exercises, should be considered together in managing the fitness of older women.

Exercise helps prevent the decline in physiological function by increasing physical strength and endurance. Additionally, sensory stimulation during postural maintenance tasks can enhance balance ability by recalibrating the weighting of functional sensory inputs [8]. Dlugosz-Bos et al. [9] reported improved balance ability in older adult women through training using Pilates equipment. Bianco et al. [10] reported improved balance assessment scores in older adults participating in ballroom dancing as a community fitness activity. These studies provide evidence of the relationship between physical activity and balance ability in older adults.

Furthermore, many studies have introduced the Instability Neuromuscular Training (INT) as an exercise method. For example, BOSU [11], Swiss Ball [12], and Wobble Boards [13] have demonstrated their effectiveness in providing surface instability. According to Picot et al. [14], strategies for ankle proprioception are dominant on firm surfaces, especially for maintaining dynamic balance in an upright posture. However, on unstable surfaces, the movement relies more on the waist due to the loss of reliability in ankle signals. Furthermore, Behm et al. [15] stated that the optimal method for increasing balance, proprioception, and spinal stability is practicing on the same surface where the skill is performed. Successful balance control during locomotion, such as gait, relies on the appropriate generation of ground reaction forces (GRF) [16]. Based on this perspective, INT on firm surfaces is considered valid as a form of dynamic stabilization training.

Another approach to creating unstable conditions is using unstable loads. This can involve utilizing the inertial of water to provide instability. Glass et al. [17,18] demonstrated the effectiveness of INT in studies utilizing water pipes, which provide rapid and irregular perturbations due to inertia changes caused by water movement. In these studies, the degree of perturbation can increase depending on hand movements, requiring compensatory muscle activation to maintain posture each time. A study by Kang et al. [19] on badminton players reported that the muscles related to knee stability were significantly more activated due to water movement in water bags during rapid deceleration than in conventional weight resistance.

Wezenbeek et al. [20] referred to water bag training as an external perturbation training method utilized by many coaches in recent years, based on dynamic perspectives and ecological approaches. The dynamic perspective posits that the interaction between environmental factors, organismic characteristics, and specific task goal is a crucial determinant of human motor control abilities [21]. Considering that falls occur unexpectedly in unforeseen situations, training for instability on rigid surfaces could potentially help individuals overcome external perturbations encountered on typical ground surface in daily life. Implementing such training programs for older women may effectively enhance their balance abilities. Research is significantly lacking regarding the application of water bag training for older adults. Given the popularity and trend of water bag training, studies are needed to generalize the training effects.

As a method for assessing the balance ability of older adults, the Timed Up and Go manual (TUG manual) test can be used to evaluate gait-related coordination. Particularly, it is highly correlated with flexion and extension function of hip joint and knee and ankle mobility [22]. The Sharpened Romberg Test (SRT) is a method for easily assessing all systems related to balance, including somatosensory, visual, and vestibular systems associated with reflexive adjustment ability of the lower limbs and static stability, without changes in position [23,24]. To assess dynamic stability, the Y-Balance Test (YBT) [25] is a highly reliable tool for injury prediction and balance identification, as it particularly evaluates ankle stability and lower limb flexibility and coordination [26].

Therefore, this study aimed to investigate the impact of INT incorporating the principle of the inertial load of water on the balance ability of healthy older women.

## 2. Materials and Methods

### 2.1. Design and Participants

The participants of this study were older adult women aged ≥ 65 years residing in B City who visited to participate in this exercise program at a senior welfare center in B city. A pre-survey determined those who had not received any medical diagnoses related to cardiovascular, metabolic, or musculoskeletal conditions and were not taking medications (such as anti-anxiety, antidepressants, or tranquilizers) that could affect the experimental results. The sample size was calculated using G*Power 3.1 Windows with an effect size set at 0.32, power at 80%, and alpha level at 0.05 [27,28]. The characteristics of the 22 participants are shown in Table 1. All participants were selected as those who could attend all 24 sessions without absence, and indeed, all participants attended every session. The recruited participants were randomly allocated into two groups, and homogeneity tests were conducted.

The homogeneity test results indicated non-significant differences in age (t = 0.682) and physical conditions (height-t = −0.643, weight-t = 0.359, BMI-t = −0.276). Furthermore, there were no significant differences between the groups. Additionally, all participants in the study confirmed that they had not engaged in any specific physical activity in the six months prior to the study. Table 1 and Figure 1 show the participants’ physical characteristics and a flow diagram of the study. All participants were informed about the study procedures, benefits, and risks, and each provided written informed consent before enrolling. This study was approved by the Research Ethics Review Committee for Research Involving Human Research Participants, Busan University of Foreign Studies (IRB no. 7001786-202212-HR-002-01), and was conducted in accordance with the guidelines of the Declaration of Helsinki.

### 2.2. Instability Neuromuscular Training Using Waterbag Program

The INT exercise program using water bags was applied to the experimental group based on a study using water bottles by Italo et al. [29]. The exercise duration and frequency were 60 min (including 10 min of warm-up, 40 min of main exercise, and 10 min of wrap-up) per session, twice a week for 12 weeks, referring to the American College of Sports Medicine (ACSM) guidelines [30]. Before starting the training, the weight was measured and the program was explained, and participants were instructed to select a weight between 3 and 5 kg (including the weight of the bag). Upon request, participants were allowed to increase the weight at their desired time. The control group followed a protocol similar to the experimental group, but they engaged in bodyweight exercises instead of using water bags. The detailed program contents are shown in Table 2 and Figure 2.

## 3. Measurements and Data Analysis

### 3.1. Gait Ability Outcomes

The TUG manual test is a highly reliable tool that allows for easy measurement of dynamic balance and mobility, such as gait [31]. The TUG manual test was validated to assess the risk of fall, and has been reported to have excellent inter- and intra-tester reliability [ICC] = 0.97–0.99) [32] and test–retest reliability (ICC = 0.99) in older adults [33]. The participants sit on a chair and, upon signal, rise from the chair and walk as quickly as possible to a point 3 m away while holding a prepared cup (filled with water up to 1 cm below the brim) in one hand. They then return to the starting point, walking backward [31]. The evaluation is based on the recorded time taken. Participants who take more than 15 s are considered at a high risk of falls. Those who take more than 30 s are evaluated at a level where independent movement is almost impossible [34] (Figure 3).

### 3.2. Dynamic Balance Outcome

The YBT is a measurement tool for evaluating the dynamic balance of the lower extremities [25]. The YBT has good interrater reliability (r = 0.84–0.91) and good content and face validity [35]. In this study, the Professional Y-Balance Test kit (Move2Perform, Evansville, IN, USA) was used for assessment (Figure 4).

The test involves extending the leg in the forward (anterior), backward to the outside (posteromedial), and backward to the inside (posterolateral) directions, measuring the distance reached. Both hands were guided to be fixed at the anterior superior iliac spine during the measurement. All participants received thorough explanations beforehand and practiced the movements twice in each direction. Additionally, a one-minute rest was taken between each measurement, and the test was conducted three times in total. The maximum distance reached was recorded as the final result. During the performance, if the supporting leg deviated or the extended leg did not return to the starting position, the trial was considered invalid and re-measured. For analysis, each participant’s leg length was measured from the anterior superior iliac spine to the midpoint of the medial malleolus. To standardize the values obtained from the YBT, the sum of all three distances was divided by three times the length of the leg and then multiplied by 100 [36].

### 3.3. Lower Limb Reactive Ability and Static Balance Outcome

The Sharpened Romberg Test (SRT) is an assessment that involves all systems related to somatosensory, vision, vestibular function, or balance [24]. The SRT requires the participant to place their dominant foot directly behind the opposite foot so that the toes of one foot touch the heel of the other, standing in a straight line. The arms are naturally crossed in front of the chest and held steady. This test demands more postural control ability than the traditional Romberg Test [37].

High test–retest reliability has been shown for the Romberg Test (eyes opened, intra class correlation coefficient [ICC] = 0.86 and eyes closed, ICC = 0.84) and Sharpened Romberg Test (eyes opened, ICC = 0.70 and eyes closed, ICC = 0.91) [38]. Participants stood near a wall to lean on if they lost stability. Timing for the eyes-open (EO) and eyes-closed (EC) conditions was measured using a stopwatch. The stopwatch was stopped and the time recorded if the participant’s posture collapsed, deviated from the line, or if they opened their eyes during the test; the maximum duration was 30 s [39] (Figure 5).

### 3.4. Data Analysis

In this study, data were analyzed using the SPSS statistical software (SPSS Statistics: v. 26, Chicago, IL, USA). The characteristics of the participants were described in terms of mean and standard deviation. The Kolmogorov–Smirnov test was used to assess the normality of the MTUG, SRT, and YBT measurements before and after the training intervention. Only SRT (EO) did not meet the assumption of normality, while all other variables did. Paired sample *t*-tests were used to examine the within-group differences in all variables before and after training, and independent sample *t*-tests were conducted between groups. Additionally, a repeated measures two-way ANOVA was performed to verify the interaction between the type of training program (group) and the period before and after training (time). The effect size was reported using partial Eta-squared values (small: <0.01; medium: 0.06; large: ≥0.14) [40]. Cohen’s d was used to calculate the effect size (small: 0.3–0.49; medium: 0.5–0.79; large: ≥0.8 [28]. All statistical significance levels were set at 0.05.

## 4. Results

The results of the TUG manual test as an indicator of gait ability, the YBT for evaluating dynamic balance, and the SRT as an indicator of lower limb reactive and static ability assessment are presented in (Table 3).

### 4.1. Gait Ability

A significant interaction between time and group was found for the TUG manual (F = 23.815, *p* < 0.001). In the within-group comparisons, a significant reduction was observed in the waterbag group between the pre-test and the 12-week post-test measurements (*p* < 0.001). In the between-group comparisons, a significant difference was noted between the pre-test and the 12-week post-test measurements (*p* < 0.05). The Cohen’s d value was 0.872, which is higher than the criterion of 0.8 for test power.

### 4.2. Dynamic Balance

A statistically significant difference was observed in the dynamic stability assessment, with a time × group interaction for the right YBT, indicated by F = 24.627, *p* < 0.001. Within-group paired sample *t*-tests revealed a significant increase from baseline to 12 weeks post-measurement in the waterbag group (*p* < 0.001), and between-group independent sample *t*-tests showed a significant increase from baseline to 12 weeks post-measurement in the waterbag group compared with the control group (*p* < 0.05), with a Cohen’s d value of 0.841, indicating a larger effect size than the criterion of 0.8.

For the left YBT, a significant difference was found in the time × group interaction, with F = 57.964, *p* < 0.001. Within-group paired sample *t*-tests indicated a significant increase from baseline to 12 weeks post-measurement in the waterbag group (*p* < 0.001), and between-group independent sample *t*-tests revealed a significant increase from baseline to 12 weeks post-measurement in the waterbag group compared with the control group (*p* < 0.05). The Cohen’s d value was 0.827, exceeding the standard criterion of 0.8 for effect size.

### 4.3. Lower Limb Reactive Ability and Static Stability

The assessment of lower limb reactive ability in a standing posture, as measured by the SRT (EO), showed no statistically significant difference in time or group interaction, with F = 0.487 and *p* = 0.493. This indicates no significant change between baseline and 12 weeks post-measurement, nor between groups.

Conversely, the SRT (EC) revealed a statistically significant difference in time and group interaction, with F = 43.114 and *p* = 0.000. Within-group paired-sample *t*-tests indicated a significant decrease from baseline to 12 weeks post-measurement in the waterbag group (*p* < 0.001), and between-group comparisons showed a significant decrease over the same time period (*p* < 0.05) when compared with the control group. The Cohen’s d value was 0.809, which is higher than the standard criterion of 0.8 for effect size.

## 5. Discussion

This study aimed to verify the positive effects of neuromuscular training using the inertial load of water in preventing falls and improving dynamic stability among older women. Aging brings about biological and functional changes, and deficits in the functioning of the nervous system can restrict the activation of motor units [41,42]. Particularly for women, postmenopausal decreases in bone density increase the risk of fractures due to osteoporosis [7], and falls can lead to such injuries, indicating the need for preventive measures. Waterbags were used to provide the inertial load of water, where the unpredictable, rapid, and irregular movements of the water acted as unexpected external perturbations to the participants. These stimuli serve as factors that disrupt dynamic balance. Upon examining the differences between the groups that trained with waterbags and that performed bodyweight training, the waterbag training group showed improvements in balance capabilities, including dynamic stability.

The TUG manual test is a fall risk indicator test that requires significant information processing abilities and dynamic stability, especially at the point of turning around while carrying a cup of water [43]. Previous studies have reported that a TUG performance time of 9.8 to 11.6 s is within the normal range, while times over 15 s indicate a higher risk of falling [31]. The TUG manual test conducted in this study involves an additional task of holding a cup of water during the typical TUG assessment. Particularly at the point of turning back, this test requires significant information processing ability and dynamic stability, making it a fall indicator [43]. The significant reduction in TUG time from before (13.18 ± 3.06) to after the intervention (9.88 ± 2.03) in the waterbag training group and indicating a significant difference between the groups in the comparison (*p* < 0.05). When analyzing human gait, it is observed that 80% of the cycle is occupied by single-leg stance, with the remaining 20% composed of double-leg support. A stable upright posture is characterized by the continuous slow shifting of weight between the legs, indicating that it is not completely static in the sagittal and coronal planes [44]. The TUG manual test requires higher levels of stabilization compared to normal walking to prevent spillage from a cup, as weight shifting occurs predominantly on one leg while carrying a waterbag. Moreover, the irregular and random movement of water within the bag provides external perturbations, aiding in single-leg support and mobility. Callisaya et al. [45] suggested that walking speed tends to decrease more easily with age in elderly women compared to elderly men, but our results demonstrated the potential to improve this aspect. Recognizing that the TUG manual test alone may be somewhat insufficient in evaluating overall dynamic stability related to falls, this study also included the YBT for assessing dynamic ability and static stability to investigate the effectiveness of this training method. Similar recent studies have also shown positive effects on walking abilities in older adults through physical activities; Sadjapang et al. [46] reported improved physical abilities and reduced TUG performance times through a combination of aerobic, resistance, and balance exercises, while Zahedian-Nasab et al. [47] found that an exercise program using game devices improved TUG performance. These findings indicate that various approaches can improve walking abilities. Furthermore, the TUG manual test requires participants to walk while holding a cup of water, suggesting that they must concentrate more to avoid spilling, adding complexity beyond mere walking.

The YBT results showed a significant increase in distance for both right and left leg support between groups (*p* < 0.05). The YBT is designed to measure balance abilities and neuromuscular control, which are essential across a wide range of disciplines. It provides a method for assessing dynamic stability and muscle strength while in a standing posture [23,48]. Although this study did not perform muscle strength tests for specific body parts, previous research suggests that, from a strength perspective, the results can be interpreted as follows: the YBT’s posterolateral reach distance suggests enhanced adductor muscle strength in the thigh, and a greater forward trunk angle for reaching leg that implies improved hamstring strength [49]. The movements in the YBT resemble a series of single-leg squats, requiring crucial hip joint functionality for stable leg extension in three directions [50]. It can be inferred that the weight of the waterbag is perceived as more unstable than its actual weight due to the inertial load of water, thus requiring more muscle activation. Ditroilo et al. [21] reported that the use of an inertial load of water in squat movements triggered higher muscle activation in the external obliques and multifidus, indicating increased trunk stability, compared with conventional methods. Calatayud et al. [51] also reported the highest muscle activation in the waterbag group in the clean and jerk training comparing a standard barbell, sand weight, and waterbag, indicating that the waterbag induces higher muscle activation. Although this study did not measure actual muscle activation through electromyography (EMG), and it is somewhat difficult to make a judgment based solely on the results, the reason for discussing the potential improvement in muscle strength is as follows. First, in this study, the control group performed movements without any load, and although some difficulty was observed in single-leg support actions, most participants appeared to adapt quickly after several repetitions. Both groups followed the same training program, but in the waterbag group, variations such as waterbag swings (swinging the bag to increase the inertia of the water) or changing the speed of moving the bag could create different variables in the same movements, leading to differences in muscle groups used and muscle activation compared with the control group. Second, a correlation exists between grip strength and balance ability. Particularly in the older group, grip strength can serve as an indicator of overall muscular strength, and it has been associated with the risk of falling along with muscular strength [52,53], indicating an association between balance ability and muscular strength. The training applied in this study required holding a shaking waterbag 5 kg in the hand, thus enhancing grip strength, ultimately satisfying the factors affecting balance ability sufficiently. In subsequent studies, there is a need for verification through actual electromyography measurements and muscle strength tests.

The SRT is a method to assess proprioception of both legs by maintaining an upright posture with eyes closed, thus evaluating balance [54], and according to Marciniak et al. [55], it is related to improvements in muscle strength and endurance. In the SRT results, there was no significant difference between the two groups in the test conducted with eyes open. Berg et al. [56] demonstrated that approximately 20% of falls occur in the evening and night, with the majority occurring between 9 p.m. and 7 a.m., suggesting that posture maintenance may be easier when visual information is clear compared to when it is not. As evidence, in the test conducted with eyes closed, the waterbag training group showed a significant increase in posture maintenance time (*p* < 0.05). Visual cues, along with the vestibular system, play a role in providing information about vertical orientation [57], so the ability to maintain posture for a longer duration in tasks where vision is blocked is presumed to be the result of effective utilization of sensory systems such as the vestibular apparatus. If the sensory input from visual information is impaired and an external perturbation is detected, other proprioception compensates for this loss [57,58]. In this study, participants were required to maintain their posture while responding to the movement of water in an unstable environment. Considering Gibson’s proposition that environmental and individual characteristics interact to determine the responses executed under given conditions [59], these findings can be regarded as a manifestation of the combined effects of environmental and individual characteristics. The improvement in postural maintenance ability was achieved through participants’ self-organization skills, with the ultimate goal of enhancing motor control ability through neuromuscular training. It is analyzed that waterbag training contributed to achieving these objectives. In previous research on older adults, changes in neuromuscular performance, as well as muscle strength, muscular endurance, bone density, and balance abilities, have been adopted as variables, with positive changes reported through various exercise interventions. Among these, studies have shown that programs using whole-body vibration devices, which require quick responses, can reduce the risk of falls among the elderly. Whole-body vibration devices increase the gravitational load on neuromuscular systems, causing muscles and tendons to detect vibrations and respond adaptively, thereby inducing reflexive muscle contractions and improving neuromuscular function [60]. Furthermore, the effectiveness of whole-body vibration training for individuals over 65 has been demonstrated through improvements in muscle strength, prevention of joint mobility decrease, and reduction in the frequency of falls [61,62], showing it to be an effective method for enhancing neuromuscular function. However, there are environmental limitations such as the need to purchase whole-body vibration equipment or access to gyms where they are available, and expertise on how to use the equipment and its effects is also required. Thus, alternative neuromuscular training methods are needed that are more accessible and cost-effective to overcome these challenges.

Nairn et al. [63] reported that training with the inertial load of water increased the activity of the abdominal muscles and the erector spinae muscle group, contributing to muscle stability. Italo et al. [29] reported improvements in walking ability and exercise capacity in a study developing an exercise program for elderly women using water containers. As seen in these studies, training programs for the elderly that utilize the inertia of water are effective interventions that can safely improve the neuromuscular system’s response capabilities without difficulty or high costs, thereby enhancing dynamic stability related to walking ability and reducing fall risk factors. The characteristics of the waterbag, being made of fabric with handles, allow easy use for older individuals who may not be accustomed to weightlifting or may have reservations about traditional exercise equipment. This accessibility is considered a positive aspect of the waterbag’s effectiveness.

## 6. Conclusions

The investigation of the effects of neuromuscular training using the inertial load of water on balance abilities and dynamic stability in women over 65 has led to the following conclusions.

After 12 weeks of training, there was a significant decrease in the task completion time associated with an increase in walking speed in the TUG manual test while carrying water cup, indicating improved function related to walking. In the YBT, an increase in distance in all directions was observed, indicating improved single-leg support ability. In the SRT, no significant difference was found in the eyes-open condition before and after the exercise, but a significant increase in posture maintenance time was observed in the eyes-closed condition in the exercise group, indicating improved lower limb reactivity related to balance. These results confirm that neuromuscular training using the inertial load of water can develop the somatosensory system and walking function related to dynamic stability in elderly women and improve balance abilities. However, the small sample size of this study presents challenges in generalizing the results. Furthermore, this study did not compare with traditional weights, subsequent research should conduct comparative studies to verify the effects based on the characteristics of water ’s inertial. Therefore, subsequent research should consider a larger sample size and more detailed factors, including gender.

## Figures and Tables

**Figure 1 jfmk-09-00050-f001:**
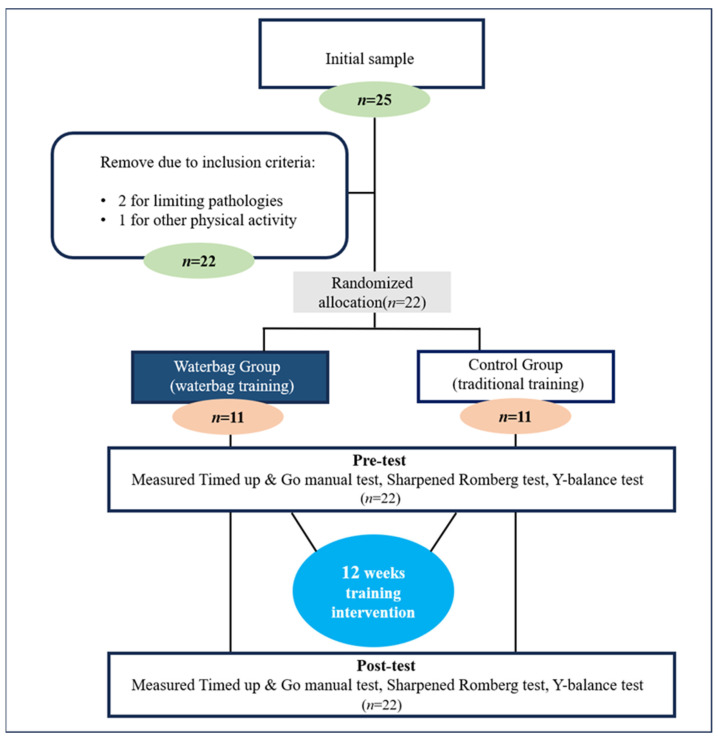
Flow diagram of the study.

**Figure 2 jfmk-09-00050-f002:**
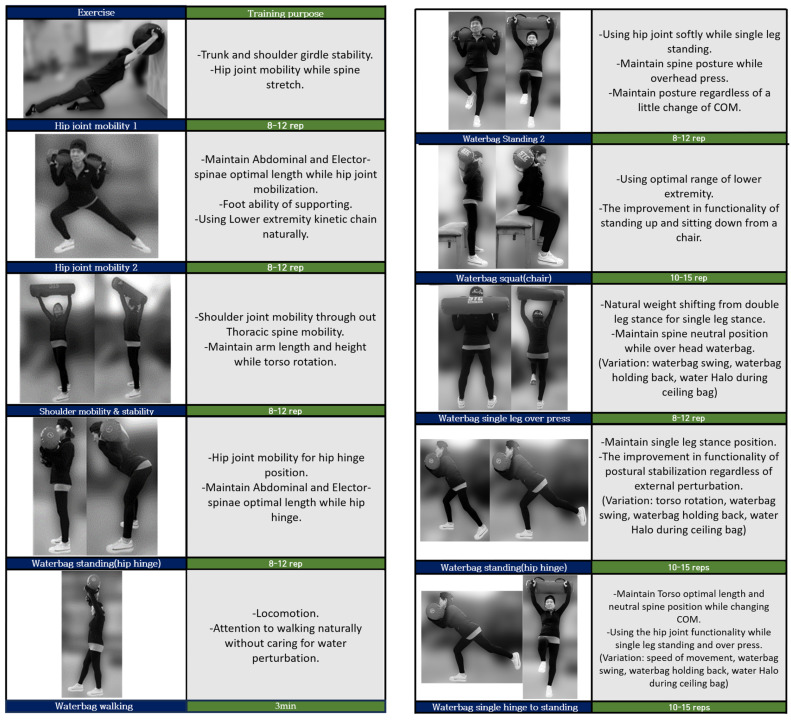
INT using water training program.

**Figure 3 jfmk-09-00050-f003:**
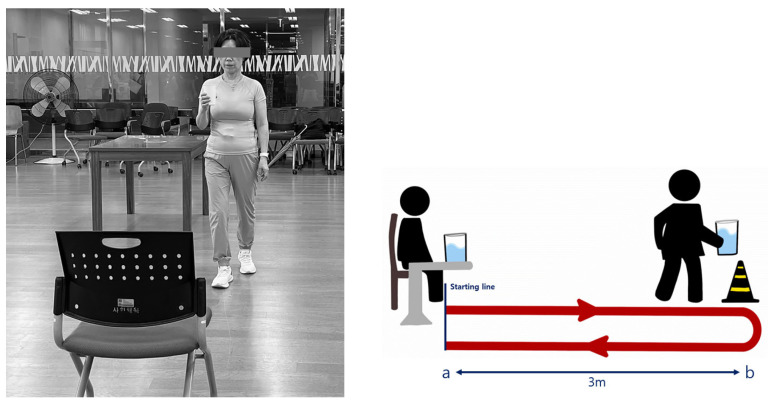
Timed Up and Go manual test.

**Figure 4 jfmk-09-00050-f004:**
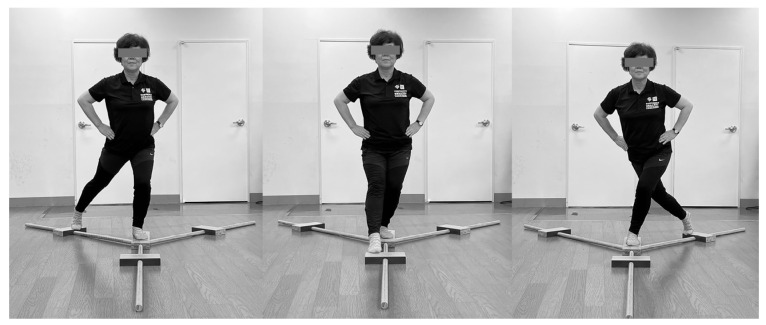
Y-balance test (**left**: posterolateral; **center**: anterior; **right**: posteromedial).

**Figure 5 jfmk-09-00050-f005:**
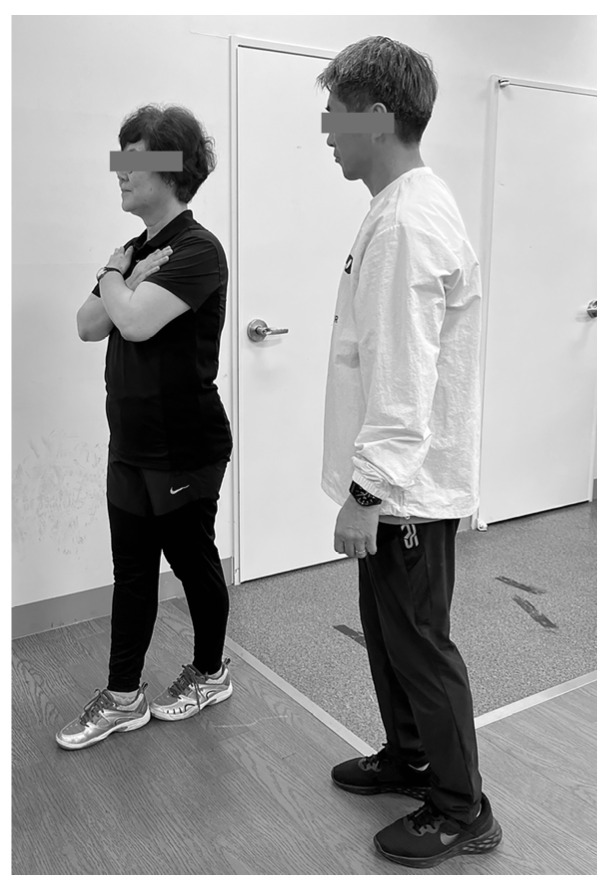
Sharpened Romberg test.

**Table 1 jfmk-09-00050-t001:** Participant’s physical characteristics (*n* = 22).

	Age (Year)	Height (cm)	Weight (kg)	BMI (kg/m^2^)
WG (*n* = 11)	74.91 ± 6.33	155.91 ± 6.35	55.13 ± 7.50	22.73 ± 3.37
CG (*n* = 11)	74.27 ± 7.92	152.49 ± 4.06	50.55 ± 7.72	24.37 ± 3.75

WG, waterbag group; CG, control group.

**Table 2 jfmk-09-00050-t002:** Instability neuromuscular training using waterbag program.

Exercise	Program	Time
Warm-up	Dynamic stretching	10 min
	1. Hip joint mobility with waterbag 1	
	2. Hip joint mobility with waterbag 2	
	3. Shoulder mobility and stability with waterbag	
	4. Waterbag standing (hip hinge) 1	
Main	5. Free walking while carrying waterbag	40 min
	6. Waterbag standing 2	
	7. Waterbag squat (chair)	
	8. Waterbag single leg over press	
	9. Waterbag standing (hip hinge) 2	
	10. Waterbag single-leg hinge to standing	
Cool-down	Static stretching and free walking	10 min

**Table 3 jfmk-09-00050-t003:** 12 weeks exercise change in balance ability.

Variables	Waterbag Group (*n* = 11)			Control Group (*n* = 11)			Group × Time ^#^		
	Pre	12 Weeks	ICC (2.1)	Pre	12 Weeks	ICC (2.1)	F	*p*-Value	η_p_^2^
MTUG	13.69 ± 3.09	9.88 ± 2.03 ^a,b^	0.118 ***	12.58 ± 2.13	11.98 ± 2.34	0.921 ***	23.815	0.000	0.544
SRT (EO)	28.45 ± 2.69	29.54 ± 1.50	0.894 ***	28.29 ± 2.64	28.63 ± 2.33	0.948 ***	0.487	0.493	0.024
SRT (EC)	12.00 ± 7.25	16.81 ± 6.11 ^a,b^	0.57	12.60 ± 6.01	11.17 ± 5.61	0.873 ***	43.114	0.000	0.683
YBT (right)	82.02 ± 6.19	92.13 ± 8.99 ^a,b^	0.642 ***	81.25 ± 10.49	83.22 ± 10.52	0.961 ***	24.627	0.000	0.552
YBT (left)	78.57 ± 8.95	89.65 ± 11.18 ^a,b^	0.742 ***	79.04 ± 9.56	80.20 ± 9.98	0.982 ***	57.964	0.000	0.743

Note: All values are expressed as means ± standard deviations. ^#^ Analysis of two-way repeated-measures ANOVA; ^a^ = Significant difference between two groups, *p* < 0.05, ^b^ = Significant difference between Pre- and Post-training, *p* < 0.001. MTUG = Modified Timed Up and Go, SRT = Sharpened Romberg Test, YBT = Y-Balance Test, EO = Eye Open, EC = Eye Closed. *** *p* < 0.001.

## Data Availability

The data used and/or analyzed during the current study are available from the corresponding author upon reasonable request.

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
