# Peer review of "Effects of Instability Neuromuscular Training Using an Inertial Load of Water on the Balance Ability of Healthy Older Women: A Randomized Clinical Trial"

_jfmk, 2024, doi:10.3390/jfmk9010050_

Round 1
Reviewer 1 Report
Comments and Suggestions for Authors
Abstract is well written; however, I would like to suggest eliminating the numerical references regarding the division of the sample into groups and I would write 22 participants in letters
I would like to suggest that authors diversify clear words with the title to improve the visibility of the manuscript in indexing
Line 40 - Which physical activity works to prevent falls? What characteristics should it have? In 2014, Bianco et al. (Group fitness activities for the elderly: An innovative approach to reduce falls and injuries) showed that the characteristics of ballroom dancing can improve balance. Some exercises can improve balance ability should incorporate a factor of coordination. This part of the manuscript should be strengthened
The recruitment phases of the analyzed sample are very little detailed. I would suggest the authors to write the paragraph in more detail
How was the sample power calculated? If it has not been calculated I would suggest entering the post-hoc power
What is the validity and reliability of the all tests? Should be specified
The statistical analysis could be strengthened by inserting a Cohen's d to calculate the effect, furthermore a Pearson analysis between the variables could give information between the relationship of the tests
Inserting graphs along with the tables would give more clarity to the manuscript
Line 198: Rhomberg Test(SRT) ???
Author Response
< Abstract part >
- about participants
- This study was conducted with twenty-two healthy older women aged ≥ 65 (mean age: 74.82±7.00 years, height: 154.20±5.49cm, weight: 55.84±7.46kg, BMI: 23.55±3.58kg/m2). The participants were randomly allocated into two groups: a group that used water bags and a control group performing bodyweight exercises.
- (pre:Line40)Which physical activity works to prevent falls? What characteristics should it have? Some exercises can improve balance ability should incorporate a factor of coordination.
- (Line52-59) Exercise helps prevent the decline in physiological function by increasing physical strength and endurance. Additionally, sensory stimulation during postural maintenance tasks can enhance balance ability by recalibrating the weighting of functional sensory inputs [8]. Dlugosz-Bos et al. [9] reported improved balance ability in older adult women through training using Pilates equipment. Bianco et al. [10] reported improved balance assessment scores in older adults participating in ballroom dancing as a community fitness activity. These studies provide evidence of the relationship between physical activity and balance ability in older adults.
(Line79-87) Another approach to creating unstable conditions is using unstable loads. This can involve utilizing the inertial of water to provide instability. Glass et al. [20-21] demonstrated the effectiveness of RNT in studies utilizing water pipes, which provide rapid and irregular perturbations due to inertia changes caused by water movement. In these studies, the degree of perturbation can increase depending on hand movements, requiring compensatory muscle activation to maintain posture each time. A study by Kang et al. [22] on badminton players reported that the muscles related to knee stability were significantly more activated due to water movement in water bags during rapid deceleration than in conventional weight resistance.
- The recruitment phases of the analyzed sample are very little detailed. / How was the sample power calculated? If it has not been calculated, entering the post-hoc power.
- (Line 109-118) The participants of this study were older adult women aged ≥ 65 years residing in B City who visited to participate in this exercise program. A pre-survey determined those who had not received any medical diagnoses related to cardiovascular, metabolic, or musculoskeletal conditions and were not taking medications (such as anti-anxiety, antidepressants, or tranquilizers) that could affect the experimental results. The sample size was calculated using G*Power 3.1 Windows with an effect size set at 0.32, power at 80%, and alpha level at 0.05 [30-31]. The characteristics of the 22 participants are shown in Table 1. Participants who could regularly participate twice a week for 12 weeks were selected. Informed consent was obtained from all participants after the purpose and procedures of the study were thoroughly explained. The recruited participants were randomly allocated into two groups, and homogeneity tests were conducted. (continuing with Line119).
(Line 217~228), (Line 234-268) I provided additional description about a Cohen’s d.
- What is the validity and reliability of the all tests? Should be specified.
- I have revised the overall flow and content of the discussion, taking into account the points raised by the reviewers.
- The statistical analysis could be strengthened by inserting Cohen’s d calculate the effect, and information of the relationship of the tests.
(Result Part)
- 1. Gait Ability
A significant inter action between time and group was found for the TUG manual (F=23.815, P<0.001). In the within-group comparisons, a significant reduction was observed in the waterbag group between the pre-test and the 12-week post-test measurements (p<0.001). In the between-group comparisons, a significant difference was noted between the pre-test and the 12-week post-test measurements (p<0.05). The Cohen’s d value was 0.872, which is higher than the criterion of 0.8 for test power.
(I have also made revision to other tests.)
- Inserting graphs along with the tables.
- Yes, I did.
- Romberg Test(SRT)=Sharpened Romberg’s Test
- ( There was a typo. I have corrected it.)
I have made efforts to incorporate that the advice provided by the reviewer as much as possible.
I Sincerely appreciate your time and assistance, and I kindle ask for your continued support.
Thank you very much.
Warm regards,
- 17 Feb.
Shuho Kang

Reviewer 2 Report
Comments and Suggestions for Authors
Dear authors,
This is a study investigating how to improve muscle strength and stability in the elderly and improve the problem of balance decline in the elderly. I think this is an interesting and meaningful report. It is recommended that authors improve the following suggestions to improve reader convenience and manuscript visualization.
1. Introduction
It is good that the author fully explains the motivation and purpose of the study in the introduction. However, if we could re-emphasize the issues of subjects and gender, add evidence of the impact of previous relevant research on this subject, and emphasize the importance of elderly women. I believe this will help the manuscript become more focused.
For example: Line 30, Fall accidents caused by...
2.Materials and Methods
a. The author can reflect subject capital information, IRB and research process very well. However, the color of the research structure diagram of the current manuscript needs to be adjusted, which is not convenient for readers.
b. It is recommended that the author explain the experimental prescription in detail, and it would be better if the author could provide picture illustrations of the 10 actions.
3. Instruments and measurements
a. It is recommended that the author add a graphic description of the operation process in Figure 3. Timed Up & Go test. For example: use marks or arrows to mark the process from A to B.
4.Discussion
Can you further explain or explain that the subjects changed the results because of the advantages, characteristics, and functions of a certain action in the experiment?
5.Conclusions
If there are still deficiencies in this experiment or if there are other findings, the author is advised to provide additional explanations. This will help subsequent researchers follow up or further improve and perfect this study.
Best regards,
Author Response
< Introduction part >
Add evidence of the impact of previous relevant research on this subject, and emphasize the importance of elderly women.
- (Line 48-52) For women, catabolic hormone activity is higher than in men, resulting in greater muscle mass loss [6]. Furthermore, women are more susceptible to fractures due to osteoporosis, caused by a decrease in bone density after menopause [7]. Balance training, as well as strength exercises, should be considered together in managing the fitness of older women. (I have added content explaining why older women should engage in exercise.)
< Materials and Methods part >
- The color of the research structure diagram
- I did. (Line under 123~)
- Provide picture illustrations of the 10 actions
- I did. (Figure 2)
< Instruments and measurements >
Add a graphic description of the process(TUG test)
- I did. (Figure 3)
< Discussion >
Explain that the subjects changed the results
- I have revised the overall flow and content of the discussion, taking into account the points raised by the reviewers. (Please check discussion part)
< Conclusions >
Add explanations and deficiencies
- The investigation of the effects of neuromuscular training using the inertial load of water on balance abilities and dynamic stability in women over 65 has led to the following conclusions. After 12 weeks of training, the task performance speed in the TUG manual test when carrying a cup of water significantly decreased, indicating improved function related to walking. In the YBT, an increase in distance in all directions was observed, indicating improved single-leg support ability. In the SRT, no significant difference was found in the eyes-open condition before and after the exercise, but a significant increase in posture maintenance time was observed in the eyes-closed condition in the exercise group, indicating improved lower limb reactivity related to balance. These results confirm that neuromuscular training using the inertial load of water can develop the somatosensory system and walking function related to dynamic stability in elderly women and improve balance abilities. However, the small sample size of this study presents challenges in generalizing the results. Therefore, subsequent research should consider a larger sample size and more detailed factors, including gender.
I have made efforts to incorporate that the advice provided by the reviewer as much as possible.
I Sincerely appreciate your time and assistance, and I kindle ask for your continued support.
Thank you very much.
Warm regards,
- 17 Feb.
Shuho Kang

Reviewer 3 Report
Comments and Suggestions for Authors
Dear Authors;
The research is interesting, but he believes that many improvements must be made before it can be published.
Title: Must indicate the type of research carried out in it.
Summary: Indicate what it is Inertial load of water.
Line 10. Put the age and weight of the participants in parentheses, indicate their sex and whether they were healthy patients.
Line 12: Indicates whether the measurement tests are validated.
Line 13: indicate statistical analyzes carried out and with what confidence intervals
Line 21: the keywords any is not Mesh Term. Please review them for these types of terms.
Introduction
Line 34- Indicates reference
Line 44. Define this type of exercise before continuing to talk about it. Could you indicate compared to other studies in which it has already shown improvements and in which it has not?
Line 87- Indicates if there are studies on the reliability of these tests.
Materials and methods
Line 91_ indicates the sociodemographic variables of the participants and the reason why they are women.
Line 97. Include the inclusion and exclusion criteria followed.
Line 106-Indicate if the study has been prospectively registered on a public access page as clinical trials.
Table 1- Abbreviations are missing
Line 117- what training did the control group do? You must explain it carefully.
Line 162The sample size calculation is missing. It seems like an especially small sample for this type of study. Do the math and look for similar studies in balance and exercise that have found significant differences with the alpha and beta error and effect size they use.
Line 166- indicate if you performed a test to identify the normality of the variables and what they were.
Table 3 indicates the abbreviations of the table with its title and put n smallest font size.
Discussion
Line 280 should talk about the importance of them being women, if they have one and the reason
Line 284- a limitation is the sample size
Line 284- Another limitation is that the exercise of the other group is not well defined and you do not know if the only difference is the water or there is some other difference.
Line 300 indicates future lines of research
Author Response
- Title: indicate the type of research carried out in it.
- Yes. Ofcorse I will. but, Could you please clarify what kind of indication you are referring to when you mention ‘Article’ in the title…? I might have misunderstood, and I apologize for any confusion. Thank you so much.
< Summary part >
- Indicate what it is Inertial load of water.
- The reflexive responses to resist external forces and maintain posture result from the coordination between the vestibular system, muscle, tendon, and joint proprioceptors, and vision. Aging deteriorates these crucial functions, increasing the risk of falls. This study aimed to verify whether a training program with water bags, an Instability Neuromuscular training (INT) using the inertial load of water, could positively impact balance ability and dynamic stability.
- Put the age and weight of the participants in parentheses, indicate their sex and whether they were healthy participants.
- This study was conducted with twenty-two healthy older women aged ≥ 65 (mean age: 74.82±7.00 years, height: 154.20±5.49cm, weight: 55.84±7.46kg, BMI: 23.55±3.58kg/m2).
- Indicates whether the measurement tests are validated.
- The pre-and post-tests were compared using t-tests to examine within- and between-group differences. The effect size was examined based on the interaction between group and time using a two-way repeated measures ANOVA.The Modified Timed Up and Go manual(TUG manual), Sharpened Romberg Test(SRT), and Y-balance test(YBT)s were conducted to assess dynamic stability, including gait function, static stability, and reactive ability. In comparison between groups, the waterbag training group showed a significant decrease in gait speed in the TUG manual test (p<0.05), an increase in static stability and reaction time in the Sharpened Romberg test with eyes closed (p<0.05), and an increase in single-leg stance ability in both legs in the Y-balance test (p<0.05).
- Indicate statistical analyzes carried out and with what confidence intervals.
- All statistical confidence interval levels were set 95%.
- The keywords
- Keywords: Dynamic stability; Inertial load of water; Instability neuromuscular training; Older women, Balance ability
< Introduction part >
- (pre Line34)=The sentence has been deleted due to modification.
- Indicate about this type exercise and compares to other studies.
- (Line 135) The control group followed a protocol similar to the experimental group, but they engaged in bodyweight exercises instead of using water bags.
- Indicate the reliability of tests.
5.1. Gait Ability
- A significant inter action between time and group was found for the TUG manual (F=23.815, P<0.001). In the within-group comparisons, a significant reduction was observed in the waterbag group between the pre-test and the 12-week post-test measurements (p<0.001). In the between-group comparisons, a significant difference was noted between the pre-test and the 12-week post-test measurements (p<0.05). The Cohen’s d value was 0.872, which is higher than the criterion of 0.8 for test power.
5.2. Dynamic balance
- A statistically significant difference was observed in the dynamic stability assessment, with a time × group interaction for the right YBT, indicated by F=24.627, P<0.001. Within-group paired sample t-tests revealed a significant increase from baseline to 12 weeks post-measurement in the waterbag group (p<0.001), and between-group independent sample t-tests showed a significant increase from baseline to 12 weeks post-measurement in the waterbag group compared with the control group (p<0.05), with a Cohen’s d value of 0.841, indicating a larger effect size than the criterion of 0.8.
For the left YBT, a significant difference was found in the time × group interaction, with F=57.964, p<0.001. Within-group paired sample t-tests indicated a significant increase from baseline to 12 weeks post-measurement in the waterbag group (p<0.001), and between-group independent sample t-tests revealed a significant increase from baseline to 12 weeks post-measurement in the waterbag group compared with the control group (p<0.05). The Cohen’s d value was 0.827, exceeding the standard criterion of 0.8 for effect size.
- 3. Lower Limb Reactive Ability and Static Stability
The assessment of lower limb reactive ability in a standing posture, as measured by the SRT(EO), showed no statistically significant difference in time or group interaction, with F=0.487 and p=0.493. This indicates no significant change between baseline and 12 weeks post-measurement, nor between groups.
Conversely, the SRT(EC) revealed a statistically significant difference in time and group interaction, with F=43.114 and p=0.000. Within-group paired-sample t-tests indicated a significant decrease from baseline to 12 weeks post-measurement in the waterbag group (p<0.001), and between-group comparisons showed a significant decrease over the same time period (p<0.05) when compared with the control group. The Cohen’s d value was 0.809, which is higher than the standard criterion of 0.8 for effect size.
< Materials and methods part >
- Indicate the sociodemographic variables of the participants and the reason why they are
- (Line 48~) For women, catabolic hormone activity is higher than in men, resulting in greater muscle mass loss [6]. Furthermore, women are more susceptible to fractures due to osteoporosis, caused by a decrease in bone density after menopause [7]. Balance training, as well as strength exercises, should be considered together in managing the fitness of older women.
- Include the inclusion and exclusion criteria followed.
- The participants of this study were older adult women aged ≥ 65 years residing in B City who visited to participate in this exercise program. A pre-survey determined those who had not received any medical diagnoses related to cardiovascular, metabolic, or musculoskeletal conditions and were not taking medications (such as anti-anxiety, antidepressants, or tranquilizers) that could affect the experimental results.
- Additionally, all participants in the study confirmed that had not engaged in any specific physical activity in the six months prior to the study.
- Indicate registered on a public access page as clinical trials.
- This study was approved by the Research Ethics Review Committee for Research Involving Human Research Participants, Busan University of Foreign Studies (IRB no. 7001786-202212-HR-002-01) and was conducted in accordance with the guidelines of the Declaration of Helsinki.
(But…Would you mind clarifying what else needs to be indicated, aside from the statement already provided? I apologize for ani convenience…)
- Abbreviations are missing
- Yes, I fixed it
- Explain about control group who what training did
- The control group followed a protocol similar to the experimental group, but they engaged in bodyweight exercises instead of using water bags.
- The sample size calculation is missing.
- The sample size was calculated using G*Power 3.1 Windows with an effect size set at 0.32, power at 80%, and alpha level at 0.05 [30-31]. The characteristics of the 22 participants are shown in Table 1. Participants who could regularly participate twice a week for 12 weeks were selected. Informed consent was obtained from all participants after the purpose and procedures of the study were thoroughly explained. The recruited participants were randomly allocated into two groups, and homogeneity tests were conducted. Etc.
- Indicate a test to identify the normality of the variables and what they ware.
(Data analysis part)
- The Kolmogorov-Smirnov test was used to assess the normality of the MTUG, SRT, and YBT measurements before and after the training intervention.
< Discussion & Conclusion>
- Should talk about the importance of them being women.
- Aging brings about biological, and functional changes, and deficits in the functioning of the nervous system can restrict the activation of motor units. Particularly for women, postmenopausal decreases in bone density increase the risk of fractures due to osteoporosis, and falls can lead to such injuries, indicating the need for preventive measures.
- A limitation is the sample size.
- However, the small sample size of this study presents challenges in generalizing the results. Therefore, subsequent research should consider a larger sample size and more detailed factors, including gender.
- Another limitation is that the exercise of the other group.(about exercise between waterbag and control group)
- In the SRT results, there was no significant difference between the two groups in the test conducted with eyes open. This suggests that maintaining a static posture under temporal information might not pose difficulty for healthy older adults, which aligns with the findings of Yoo(2019), where no differences were observed between groups with superior and average motor abilities in unstable posture with eyes open.
- Indicate future lines of research.
- However, there is a need to verify the correlation between the YBT results and muscle strength improvement through specific and practical strength tests, and subsequent studies are deemed necessary to include electromyography measurements and muscle strength tests for verification.
I have made efforts to incorporate that the advice provided by the reviewer as much as possible.
I Sincerely appreciate your time and assistance, and I kindle ask for your continued support.
Thank you very much.
Warm regards,
- 17 Feb.
Shuho Kang

Round 2
Reviewer 1 Report
Comments and Suggestions for Authors
The authors answered all my doubts and suggestions
Reviewer 2 Report
Comments and Suggestions for Authors
Thanks to the author's efforts, I feel that the quality of this manuscript has been improved. I recommend that the editor-in-chief adopt the revised manuscript for the next step in the process.
Best regards,Author Response
Please see the attachment.
Thank you very much.

Reviewer 3 Report
Comments and Suggestions for Authors
Dear authors,
I think the article has improved but it should include references to the improvements made in the discussion about the importance of them being women.
On the other hand, I think it is good that it indicates the type of study carried out in the title, namely clinical trial or pretest-posttest.
Could you do a reliability analysis with the ICC and the standard error of measurement?
Indicate a reference of a study that with that effect size found significant differences for this type of study on proprioceptive exercise.
Try not to sign the answer sheet again as it must be anonymous.
Thank you
